# eIF4A1 Is a Prognostic Marker and Actionable Target in Human Hepatocellular Carcinoma

**DOI:** 10.3390/ijms24032055

**Published:** 2023-01-20

**Authors:** Sara M. Steinmann, Anabel Sánchez-Martín, Elisabeth Tanzer, Antonio Cigliano, Giovanni M. Pes, Maria M. Simile, Laurent Desaubry, Jose J.G. Marin, Matthias Evert, Diego F. Calvisi

**Affiliations:** 1Experimental Tumor Pathology, Institute of Pathology, University Regensburg, Franz-Josef-Strauss-Allee 11, 93053 Regensburg, Germany; 2Experimental Hepatology and Drug Targeting (HEVEPHARM) Group, University of Salamanca, IBSAL, 37007 Salamanca, Spain; 3Department of Medicine, Surgery, and Pharmacy, University of Sassari, 07100 Sassari, Italy; 4Therapeutic Innovation Laboratory, UMR7200, CNRS/University of Strasbourg, CEDEX, 67401 Illkirch, France

**Keywords:** hepatocellular carcinoma, eIF4A1, translation inhibitors, targeted therapies

## Abstract

Hepatocellular carcinoma (HCC) is a primary liver tumor with high lethality and increasing incidence worldwide. While tumor resection or liver transplantation is effective in the early stages of the disease, the therapeutic options for advanced HCC remain limited and the benefits are temporary. Thus, novel therapeutic targets and more efficacious treatments against this deadly cancer are urgently needed. Here, we investigated the pathogenetic and therapeutic role of eukaryotic initiation factor 4A1 (eIF4A1) in this tumor type. We observed consistent eIF4A1 upregulation in HCC lesions compared with non-tumorous surrounding liver tissues. In addition, *eIF4A1* levels were negatively correlated with the prognosis of HCC patients. In HCC lines, the exposure to various eIF4A inhibitors triggered a remarkable decline in proliferation and augmented apoptosis, paralleled by the inhibition of several oncogenic pathways. Significantly, anti-growth effects were achieved at nanomolar concentrations of the eIF4A1 inhibitors and were further increased by the simultaneous administration of the pan mTOR inhibitor, Rapalink-1. In conclusion, our results highlight the pathogenetic relevance of eIF4A1 in HCC and recommend further evaluation of the potential usefulness of pharmacological combinations based on eIF4A and mTOR inhibitors in treating this aggressive tumor.

## 1. Introduction

Liver cancer is one of the most frequent malignancies and has poor prognosis, ranked as the third leading cause of cancer-related deaths worldwide and thus remains a global health issue [1]. The incidence of liver cancer is increasing, and the World Health Organization has estimated that the number of deaths from this cancer will reach one million annually in 2030 [2]. Hepatocellular carcinoma (HCC) is the most common type of primary liver cancer, accounting for ~90% of cases. HCC commonly develops in the setting of underlying liver diseases such as cirrhosis or chronic liver inflammation. Although hepatitis B or C virus infection and alcohol abuse remain major HCC risk factors, the number of HCC cases related to non-alcoholic fatty liver disease (NAFLD) or associated with metabolic syndrome or diabetes mellitus is rapidly growing, and this condition might become the leading cause of HCC in Western countries in the near future [3,4]. Surgical resection, radiofrequency ablation, transarterial chemoembolization, and liver transplantation are potentially curative treatments for early-stage HCC lesions [5,6]. However, most patients with HCC are diagnosed in an advanced stage of the disease when they are no longer good candidates for curative strategies. Palliative systemic therapy is the only therapeutic option for these patients. Several tyrosine multikinase inhibitors, including sorafenib, lenvatinib, regorafenib, and cabozantinib, are approved as the first- or second-line treatments for unresectable advanced HCC. However, only a slight increase in overall survival has been reached in these patients [7,8,9,10]. Immune-based therapies are also emerging for the treatment of advanced HCC. The combination of azetolizumab (anti-PD-L1) with bevacizumab (anti-VEGF) has shown superior efficacy compared to sorafenib, and it is now recommended as the standard first-line treatment for advanced HCC [6,11]. Nevertheless, the benefits of immunotherapy are still modest, and many patients remain non-responders. Therefore, developing novel therapeutic strategies is imperative to improving the outcome of HCC patients.

Targeted inhibition of translation machinery is emerging as a promising cancer treatment approach. Protein synthesis is a highly controlled process that plays a major role in gene expression regulation. However, this mechanism is frequently dysregulated in human malignancies [12]. Studies have shown that altered mRNA translation leads to tumorigenesis and cancer progression by selectively enhancing the synthesis of proteins involved in cell proliferation, activation of invasion and metastasis, and other neoplastic characteristics-related processes [13]. Translation initiation is the rate-limiting step; it is regulated by multiple eukaryotic initiation factors (eIFs) [14,15]. The eIF4F complex, which includes eIF4E, eIF4G, and eIF4A proteins, is essential for cap-dependent ribosome recruitment and translation initiation. Thus, eIF4E binds to the m^7^G-cap structure at the mRNA ‘5’ end, whereas eIF4G acts as a scaffold protein recruiting eIF4E and eIF4A. The latter, the only component of the eIF4F complex with enzymatic activity, is an ATP-dependent DEAD-box RNA helicase that unwinds the mRNA secondary structures in the 5′ untranslated regions (5′-UTRs) to enable ribosome scanning. The helicase eIF4A is required to efficiently translate mRNAs with long and highly structured 5′-UTRs, including those with G-quadruplex structures [16]. These complex structures characterize the 5-UTRs of many oncogenes considered eIF4A-dependent genes, such as KRAS, BLC2, NOTCH1, CDK6, and CCND1 [17,18].

There are two eIF4A isoforms involved in mRNA translation, eIF4A1 and eIF4A2, which share ~90% homology in their amino acid sequence [19]. Despite their similarity, eIF4A1, but not eIF4A2, is essential to cell viability. In addition, eIF4A2 cannot compensate for eIF4A1 dysfunction, suggesting that both isoforms have distinct biological properties [20,21]. Their expression also varies between different types of tissues, with eIF4A1 being the most abundant paralog [22]. The overexpression of the eIF4A1 isoform occurs in a wide variety of malignancies, where it correlates with metastasis and poor prognoses [23,24,25]. The c-Myc protooncogene upregulates EIF4A1 transcription [26]; in addition, the levels of free functional eIF4A1 protein are increased by the degradation of the tumor suppressor programmed cell death 4 (PDCD4), which is itself regulated by mammalian target of rapamycin (mTOR) signaling [27,28]. Although little is known about eIF4A2 in cancer, recent studies have revealed the upregulation of this isoform in colorectal cancer [29] and paclitaxel-resistant breast cancer [30]. High eIF4A2 levels are also associated with poor prognosis in colorectal and esophageal squamous cell carcinomas [29,31].

Consequently, the characteristics of eIF4A isoforms make them attractive drug targets for anti-tumor therapy. Rocaglates, also known as flavaglines, are a group of small molecules harboring a common cyclopean[b]benzofuran core that suppresses cap-dependent translation by inhibiting eIF4A activity. More than one hundred natural rocaglates have been isolated from *Aglaia* species, including silvestrol and rocaglamide. In addition, numerous synthetic derivatives have recently been developed to improve their potency and bioavailability [32,33,34]. The rocaglates stabilize the non-specific binding of eIF4A to RNA, which prevents its incorporation into the eIF4F complex [35,36], and they target both eIF4A isoforms [37]. Silvestrol, one of the best-studied drugs of this family, has demonstrated potent anti-tumor activity both in vitro and in vivo [38,39], and the antiproliferative potency of other active rocaglates derivatives, such as the synthetic compound CR-1-31-B, has also been reported in several tumors [40,41,42,43]. Similarly, the synthetic flavagline FL3 exhibited anticancer properties in various experimental cancer models [44]. Furthermore, the synthetic eIF4A inhibitor, Zotatifin, is currently being evaluated in a phase I/II clinical trial for solid tumors [45].

In this study, we investigated the pathogenetic role of eIF4A1 in human HCC and its potential usefulness as a novel molecular target for the development of novel pharmacological strategies.

## 2. Results

### 2.1. eIF4A1 Is Significantly Overexpressed in Human Hepatocellular Carcinoma and Correlates with a Worse Outcome

First, *eIF4A1* mRNA levels in human cancers were investigated using data obtained from The Cancer Genome Atlas (TCGA) and the UALCAN (http://ualcan.path.uab.edu/; accessed on 24 November 2022) analysis tool. *eIF4A1* mRNA expression was elevated in various cancer types, including colon adenocarcinoma (COAD), glioblastoma multiforme (GBM), esophageal carcinoma (ESCA), and kidney renal clear cell carcinoma (KIRC), among others (Figure 1A). In HCC (normal tissue: *n* = 50; tumor tissue: *n* = 371), *eIF4A1* was significantly overexpressed (Figure 1A; LIHC: liver hepatocellular carcinoma; Figure 1B; P = 8.871× 10^−10^) with comparable incidences for women (P = 2.566 × 10^−10^) and men (P = 2.155 × 10^−10^; Figure 1C) and elevated expression for patients aged between 21 and 80 years (Figure 1D; 21–40 years: P = 1.463 × 10^−06^; 41–60 years: P = 2.841 × 10^−08^; 61–80 years: P = 7.070 × 10^−09^) compared to that in normal tissue. Maximal *eIF4A1* expression was detected in tumors of young patients between 21 and 40 years of age (Figure 1D; 21–40 vs. 61–80 years: P = 2.678 × 10^−02^). Examination of multiple clinicopathological features revealed that HCC patients with high *eIF4A1* expression had poorer prognoses. The survival time of patients with high *eIF4A1* expression was significantly shorter than that of patients with low or medium *eIF4A1* levels (Figure 1E; P = 0.027). Furthermore, the TCGA data showed that in comparison to normal tissue, *eIF4A1* expression was increased in all tumor grades, with grades 3 and 4 tumors revealing even higher levels than grades 1 and 2 tumors (Figure 1F; Normal vs. Grade 1: P = 2.359 × 10^−03^; Normal vs. Grade 2: P = 3.212 × 10^−09^; Normal vs. Grade 3: P = 4.581 × 10^−09^; Normal vs. Grade 4: P = 6.219 × 10^−03^; Grade 1 vs. Grade 3: P = 5.378 × 10^−04^; Grade 1 vs. Grade 4: P = 3.380 × 10^−02^; Grade 2 vs. Grade 3: P = 2.050 × 10^−02^; Grade 2 vs. Grade 4: P = 2.020 × 10^−02^). The analysis of HCC cancer stages (displayed significantly increased *eIF4A1* expression in stage 1 to 3 tumors (Figure 1G; Normal vs. Stage 1: P = 1.878 × 10^−08^; Normal vs. Stage 2: P = 3.364 × 10^−07^; Normal vs. Stage 3: P = 1.058 × 10^−06^; Stage 1 vs. Stage 3: P = 2.366 × 10^−02^). Importantly, high *eIF4A1* expression was found in N0 and N1 nodal metastasis status (Figure 1H; Normal vs. N0: P = 4.421 × 10^−12^; Normal vs. N1: P = 1.023 × 10^−04^), with a clear tendency toward enhanced *eIF4A1* expression in N1 specimens.

To confirm the findings in the TCGA dataset analysis, the eIF4A1 levels in 47 human HCC samples and paired normal tissues collected at the University Regensburg were evaluated. In compliance with the TCGA data set, western blot analysis revealed that eIF4A1 protein levels were significantly increased in HCC specimens compared to those in paired non-tumorous tissues (Figure 2; *** *p* = 0.0002; Wilcoxon test; Appendix A). In addition, immunoblotting showed augmented protein expression of the eukaryotic initiation factor (eIF) family member eIF4A2 (* *p* = 0.0249; Wilcoxon test) and the translation initiator and rocaglate target DEAD-box protein 3 DDX3 [46] (** *p* = 0.0088; Wilcoxon test) in HCC lesions (Figure 2A,B; Appendix A).

Remarkably, when assessing the prognostic relevance of the three proteins, eIF4A1 levels were inversely associated with patient survival time (*p* < 0.0001; Figure 3A; Appendix A). In contrast, levels of eIF4A2 did not separate HCC patients based on survival length (P = 0.741; Figure 3B; Appendix A). Moreover, DDX3 displayed a trend toward more prolonged survival without reaching significance (P = 0.142; Figure 3C; Appendix A).

Subsequently, we evaluated the levels of eIF4A1 in a large collection of formalin-fixed, paraffin-embedded HCC samples (n = 356) by immunohistochemistry (Figure 4). In agreement with western blot data, increased immunoreactivity for eIF4A1 was detected ubiquitously in HCC lesions compared to that in the matching non-neoplastic liver tissues (Figure 4, first lower panel). In contrast, non-tumorous epithelial cells exhibited faint or absent eIF4A1 staining (Figure 4, second lower panel). HCC lesions displayed either homogeneous or scattered upregulation of eIF4A1 (Figure 4, third to fifth lower panels).

Overall, the present data indicated that eIF4A1 is highly expressed in primary HCC lesions and associated with a poor outcome. These findings suggest the usefulness of eIF4A1 as a prognostic biomarker in human HCC.

### 2.2. Targeting eIF4A1 with Rocaglates Inhibits Tumor Cell Growth in HCC Cell Lines

Next, to evaluate the therapeutic potential of targeting eIF4A1 activity in HCC, we challenged human HCC cell lines in vitro with five distinct rocaglate derivatives, namely CR-1-31-B, FL3, Rocaglamide, Silvestrol, and Zotatifin. These drugs promote the binding affinity of eIF4A1 to mRNA species containing polypurine sequence motifs, sequentially impeding the translation initiation during protein synthesis by blocking the formation of the heterotrimeric eIF4F cap-binding complex. Prior to pharmacological treatments, we assessed the endogenous eIF4A1, eIF4A2, and DDX3 protein levels using western blot analysis in various human HCC cell lines. eIF4A1, eIF4A2, and DDX3 proteins were expressed in all HCC cell lines tested (Figure 5A,B; Appendix A).

Subsequently, to assess the cytotoxic potential of eIF4A inhibitors, the HLE, HLF, and PLC/PRF/5 tumor cell lines were randomly selected and treated with a range of concentrations between 0 and 200 nM per eIF4A inhibitor, and colorimetric MTT assays were performed 48 h after starting compound exposure. Principally, cellular metabolic activity was measured to indicate cell viability, proliferation, and cytotoxicity. Overall, all five rocaglates efficiently reduced tumor cell viability in the three cell lines in a dose-dependent manner (Figure 6A–C). Notably, in HCC cell lines, the calculated half-maximal inhibitory concentration (IC50) values for the five drugs resided in the low nanomolar range (Figure 6D).

To further elucidate the mechanisms of action of eIF4A inhibitors in HCC cell lines, we selected the two most effective drugs, i.e., CR-1-31-B and Zotatifin, for the following experiments. Zotatifin was also selected because, to date, it is the only eIF4A inhibitor in clinical trials. When evaluating cell proliferation, we found that incubation with CR-1-31-B and Zotatifin induced a marked reduction in BrdU incorporation in both HLE and HLF cell lines, with a slightly more pronounced anti-growth effect in the case of CR-1-31-B (Figure 7A,B).

Subsequently, apoptosis was determined in the same cell lines treated with the two eIF4A inhibitors. Both CR-1-31-B and Zotatifin induced higher apoptotic cell death in the cell lines than was observed in untreated and DMSO-treated cells (Figure 8A,B).

Next, we assessed the effects of CR-1-31-B and Zotatifin on some of the most relevant oncogenic pathways in hepatocarcinogenesis by western blot analysis (Figure 9; Appendix A). Both drugs did not affect eIF4A1 and DDX3 protein levels, supporting the hypothesis that they act on eIF4A1 and DDX3 activity without affecting their expression. In contrast, similarly to what has been described in other tumor types, the treatment of HCC cells with these drugs led to eIF4A2 upregulation, which is considered a compensatory response to marked eIF4A1 suppression. Nonetheless, it has been shown that eIF4A2 induction cannot functionally compensate for eIF4A1 loss or inactivation [20,21]. Regarding oncogenic pathways, both CR-1-31-B and Zotatifin administration resulted in the downregulation of activated/phosphorylated (p-)STAT3, AKT, ERK1/2, and total SKP2 proteins in a dose-dependent manner. In addition, levels of the Hippo pathway effector protein TAZ were decreased, whereas the effects on the ortholog YAP were inconsistent. Furthermore, the levels of prohibitin 1 and 2 (PHB1 and 2), which are supposed to be critical targets of rocaglates [47], were decreased in CR-1-31-B- and Zotatifin-treated cells. In contrast, levels of mTOR effectors, such as activated/phosphorylated (p-)RPS6 and phosphorylated/inactivated (p-)4EBP1, were either increased or unchanged following CR-1-B-31 and Zotatifin administration.

### 2.3. Targeting the mTOR Pathway Synergizes with eIF4A Inhibitors to Restrain HCC Cell Growth In Vitro

Inspired by the findings that eIF4A inhibitors did not reduce the activation of mTOR effectors, we tested whether mTOR suppression could synergize with eIF4A inhibitors to hamper HCC cell growth. For this purpose, the HLE and HLF cell lines were treated with the pan mTOR inhibitor, Rapalink-1 [48], either alone or in combination with CR-1-31-B or Zotatifin. Rapalink-1 was chosen over common mTORC1 inhibitors, such as Rapamycin or Rapamycin homologs, because the latter drugs effectively inhibit RPS6 but not 4EBP1 [49]. When assessing HLE and HLF metabolic activity with the MTT assay, we discovered that Rapalink-1 inhibited HCC cell growth at nanomolar concentrations (IC50: 20 nM). Significantly, the combination of Rapalink-1 with CR-1-31-B or Zotatifin triggered an even more potent, synergistic reduction of HLE and HLF metabolic activity than that induced by treatments alone (Figure 10).

Subsequently, we assessed proliferation and apoptosis in the same cell lines using Rapalink-1 alone and in combination with CR-1-B-R1 or Zotatifin. Again, the combined treatments resulted in a significant reduction in proliferation (Figure 11A) and a robust increase in apoptosis (Figure 11B) in the two cell lines.

Finally, western blot analysis confirmed that the mTOR inhibitor, Rapalink-1, alone and in combination with the eIF4A inhibitors, CR-1-31-B or Zotatifin, fully inhibited p-RPS6 and p-4EBP1 levels. In addition, Rapalink-1 alone resulted in the downregulation of eIF4A1, eIF4A2, and DDX3, whereas inconsistent results were obtained for these proteins when Rapalink-1 was combined with CR-1-31-B or Zotatifin (Figure 12; Appendix A). These data indicated that targeting the eIF4A family and mTOR oncogenic pathway might be a potent anti-tumor treatment strategy in HCC cell lines (Figure 12; Appendix A).

## 3. Discussion

HCC, the most common liver primary cancer, is characterized by its late diagnosis, increasing incidence worldwide, and limited treatment options [51]. Despite the most recent and effective therapeutic strategies [11,52,53,54], the prognosis for advanced HCC remains poor [55]. Therefore, the identification of novel molecular targets and development of more efficacious therapies are imperative to improving the life expectancy of HCC patients.

Cumulating evidence indicates that translation dysregulation is critical to tumor development and progression [56]. Therefore, targeting translation initiation components might be a potential strategy for therapeutic interventions against cancer [57]. In the present study, we investigated in silico and in vitro the pathogenetic and therapeutic relevance of eIF4A1, a pivotal player in initiating cap-dependent protein translation, in HCC [19,57]. Our results strongly suggest a pro-tumorigenic function of eIF4A1. Specifically, we show that HCC patients with high levels of eIF4A1 mRNA and protein carry poor outcomes. Thus, eIF4A1 might represent a novel prognostic marker in this disease. Accordingly, a negative prognostic role of eIF4A1 has been previously detected in other tumor types, such as lung adenocarcinoma [58], clear cell renal carcinoma [59], and gastric cancer [23], among many others.

Furthermore, we revealed that targeting eIF4A1 using rocaglates induced strong growth suppression of human HCC cells by inhibiting proliferation and triggering apoptosis. Notably, the growth inhibitory activity of these compounds on HCC cells occurred at low nanomolar concentrations, implying their potency, at least in vitro. These data, confirming previous findings obtained in different tumor types, support the targeting of eIF4A1 as a promising therapeutic strategy against HCC [57].

At the molecular level, we discovered that various oncoproteins belonging to the JAK/STAT, ERK/MAPK, AKT, and Hippo/TAZ pathways are markedly downregulated by eIF4A inhibitors. On the other hand, we revealed that treatment with these compounds did not affect the activity of members of the mTOR signaling cascade. Strikingly, exposure of HCC cells to combinations of eIF4A inhibitors and the mTOR inhibitor, Rapalink-1, a third-generation mTOR inhibitor that links rapamycin and MLN0128 [48], resulted in synergistic cytostatic effects. Indeed, this treatment led to higher suppression of proliferation and induction of massive apoptosis compared to that induced by incubation with one of the compounds alone. To the best of our knowledge, this is the first report showing the synergistic effects of eIF4A and mTOR inhibitors against the growth of HCC cells. Furthermore, preliminary data from our laboratory indicate that this drug combination is also synergistic in other tumor entities in vitro, suggesting that cancer cells might be highly vulnerable to eIF4A and mTOR pathway inhibition.

Moreover, this is the first study to demonstrate in the setting of HCC that Rapalink-1-induced inhibition of mTOR signaling can be achieved through suppression of two key downstream substrates, p-RPS6^Ser235/236^ and p-4EBP1^Thr37/46^, even in the presence of eIF4A inhibitors. Hence, we suggest that a combinatory therapy with eIF4A inhibitors and Rapalink-1, a drug that was originally developed to overcome mTOR resistance mutations, might serve as a refinement strategy for systemic management of HCC by simultaneously and effectively targeting both mTOR signaling and eIF4A-dependent translation. Overall, the present data indicate that eIF4A is a potential prognostic biomarker in HCC patients and a promising therapeutic target in this deadly tumor type. In particular, eIF4A inhibitors displayed formidable cytostatic activity against HCC cells in vitro, further augmented in combination with an mTOR inhibitor. Although the antineoplastic effectiveness of these drugs, alone and in combination, requires validation in vivo, the present data suggest that targeting translation initiation can provide an innovative and valuable strategy in the development of novel treatments against human HCC.

## 4. Materials and Methods

### 4.1. Human Tissue Specimens

Forty-seven human HCC tumor tissue samples and paired surrounding non-tumorous tissues were collected at the Institute of Pathology at the University of Regensburg (Regensburg, Germany). Patient features are summarized in Appendix A. HCC tumors were divided between shorter survival/poorer outcome (HCCP; *n* = 25) and longer survival/better outcome (HCCB; *n* = 22), characterized by <3 and >3 years’ survival following partial liver resection, respectively. The study was conducted according to the guidelines of the Declaration of Helsinki and approved by the Clinical Research Ethics Committee of the Medical University of Regensburg (protocol code 17-1015-101; 4 July 2018).

### 4.2. Cell Lines and Reagents

The human HCC cell lines HLE (JCRB0404, Xenotech, Kansas City, MO, USA), HLF (JCRB0405, Tebu-bio GmbH, Offenbach, Germany), PLC/PRF/5 (300315, CLS Cell lines Service GmbH, Eppelheim, Germany), SNU182 (CRL-2235, LGC Standards GmbH), and SNU449 (CRL-2234, LGC Standards GmbH) were cultured at 37 °C in a 5% CO_2_ humidified atmosphere. HLE, HLF, and PLC/PRF/5 cell lines were maintained in Dulbecco’s Modified Eagle Medium (DMEM, high glucose, Anprotec, Bruckberg, Germany), and SNU182 and SNU449 cell lines were maintained in RPMI 1640 (Anprotec) supplemented with 10% (*v/v*) FCS (Anprotec, Bruckberg, Germany), 2 mM L-glutamine, 10 mM HEPES, 1 mM sodium pyruvate, and 1% penicillin/streptomycin solution (all from Anprotec). Mycoplasma-free status for all cell lines was monitored regularly using the PCR Mycoplasma Test Kit I (PK-CA91-1096, PromoCell, Heidelberg, Germany), and cell line authentication was performed by Cell Lines Service (Eppelheim, Germany). CR-1-31-B (HY-136453, MedChemExpress, Cas. No. 1352914-52-3), FL3 [60], Rapalink-1 (HY-111373, MedChemExpress, Cas. No. 1887095-82-0), Rocaglamide (HY-19356, MedChemExpress, Cas. No. 84573-16-0), Silvestrol (HY-13251, MCE, Cas. No. 697235-38-4), and Zotatifin (HY-112163, MedChemExpress, Cas. No. 2098191-53-6) were used for in vitro experiments. Stock solutions (1 and 10 mM) were prepared in dimethylsulfoxide (DMSO) and aliquots were stored at −20 °C.

### 4.3. MTT Viability, Proliferation, and Apoptosis Assays

Cells were seeded in flat-bottom 96 well plates at a density of 10 x 10^4^ cells per well. After an overnight attachment period, cells were exposed to various concentrations of CR-1-31-B, FL3, Rocaglamide, Silvestrol, Zotatifin, and Rapalink-1 alone or CR-1-31-B-Rapalink-1 and Zotatifin-Rapalink-1 combinations for 48 h. Cells treated with DMSO and wells containing only culture medium served as negative and background controls, respectively.

For the MTT assay, following treatment, 10 µL of 5 mg/mL methyl-thiazolyl-diphenyl-tetrazolium bromide (MTT) solution was added per 96 well and incubated at 37 °C and under 5% CO_2_ for 2 h. The medium was fully aspirated, and 100 µL of 100% (*v/v*) DMSO was added per well to dissolve the formazan crystals. After 15 min of shaking at room temperature, the absorbances were measured at 570 and 630 nm using the FLUORstar Omega multiplate reader and MARS data analysis software (both from BMG Labtech, Ortenberg, Germany). All experiments were performed in triplicate or quadruplicate and repeated at least twice. The average absorbance of the DMSO-treated cells was defined as 100% cell viability. IC50 values and standard deviations were calculated using GraphPad Prism software. Average cell viability values of three independent MTT assay experiments were employed to determine dose-effect curves (Fa = fraction affected) and the combination indices of two drugs using CompuSyn PC software version 1 (ComboSyn Inc., Paramus, NJ, USA).

Cell proliferation was evaluated in the cell lines at the 48-h time point using the BrdU Cell Proliferation Assay Kit (Cell Signaling Technology, Danvers, MA, USA). Briefly, following drug treatment, cells were incubated with 1× bromodeoxyuridine (BrdU) for 2 h and fixed for 30 min at room temperature. The fixing solution was discarded, and cells were incubated with the BrdU mouse detection antibody for 1 h at room temperature. After washing, cells were stained with HRP-conjugated anti-mouse secondary antibody for 30 min at room temperature and washed again. After incubation with TMB substrate solution for a further 30 min at room temperature, stop solution was added, and the absorbance was measured at 450 nm. The results are expressed as fold-change over DMSO control.

Apoptosis was determined in the HCC cell lines using the Cell Death Detection Elisa plus Kit (Roche Molecular Biochemicals, Indianapolis, IN, USA), following the manufacturer’s instructions. All cell line experiments were repeated at least three times in triplicate.

### 4.4. Protein Extraction, Western Blot Analysis, and Smart Protein Layers (SPL) Approach

Human HCC tumor and non-tumorous liver tissues were homogenized in 100–250 µL of T-PER^TM^ lysis buffer (78510, Thermo Fisher Scientific) containing 1x Halt^TM^ Protease and Phosphatase Inhibitor Cocktail (78443, Thermo Scientific) in the Bullet Blender Storm 24 (Next Advance, Troy, NY, USA) for 3 min at speed 8, placed in the rotator mixer for 30–60 min at +4 °C and sonicated. Lysis of pellets from the cell lines was performed in the same lysis buffer for 1 h on ice. Protein concentrations were determined using the colorimetric BioRad Protein Assay Dye Reagent Concentrate (500-0006, Bio-Rad, Hercules, CA, USA) with bovine serum albumin (BSA) as the standard.

For western blot analysis, protein lysates were denatured in 1× LDS Bolt^TM^ sample buffer (B0007) and 1× Bolt^TM^ reducing agent (B00009, both from Invitrogen) by boiling for 5 min at 95 °C. Samples with 10 µL of protein per lane were separated by SDS-PAGE on Bolt™ 4–12% Bis-Tris Mini Protein Gels (NW04125BOX) in 1× Bolt™ MES SDS Running Buffer (B000202, both from Invitrogen) and transferred onto nitrocellulose membranes (iBlot^TM^ 2 Mini/Regular Transfer Stacks, IB23001/2) by electroblotting running program P0 in the iBlot™ 2 Gel Transfer Device (IB21001, both from Thermo Fisher Scientific). Membranes were blocked in EveryBlot Blocking Buffer (12010020, Bio-Rad) and probed at +4°C overnight with specific antibodies: eIF4A1 (1:1000; ab31217, Abcam, Cambridge, UK); eIF4A2 (1:1000; ab31218, Abcam); DDX3 (1:1000; A300-474A, Bethyl Laboratories, Montgomery, USA); p-RPS6^Ser235//236^ (4856), p-4EBP1^Thr3747^ (2885), p-STAT3^Tyr705^ (9145), AKT (4691),), ERK1/2 (4695), p-ERK1/2^Thr202/204^ (4370), SKP2 (2652), YAP/TAZ (8418), p-YAP/TAZ (13008), PHB1 (2426), PHB2 (14085) (all 1:1000; Cell Signaling Technology); p-AKT^Ser473^ (1:1000; 66444-1-IG, Proteintech (Rosemont, IL, USA); and β-actin (1:5000; ab20272, Abcam). Horseradish peroxidase-conjugated secondary antibody (1:20,000) was incubated for 1 h at room temperature, and blots were imaged with Clarity Max ECL Western Blotting Substrate (1705062S, Bio-Rad) using ChemiDoc MP (17001402, Bio-Rad). Band volumes were quantified with Image Lab^TM^ software version 6.0.1 (Bio-Rad) and normalized to GAPDH or β-actin.

High patient-to-patient variation of reference protein expression, such as GAPDH and β-actin, impedes proper quantification of western blot data. Hence, for accurate target protein quantification of human HCC samples, western blotting was performed using the Smart Protein Layers (SPL) Kit Red from NH DyeAgnostics (PR913, Halle, Germany) according to the manufacturer’s instructions. The SPL technology is based on three components: (i) red-fluorescent Smart Label (SMA label) reagent for labeling and visualization of total protein on gels and blots, (ii) bi-fluorescent sample-specific standard Smartalyzer L (SMA basic L) for normalization, standardization, and quantification of total protein per sample, and (iii) a calibrator (CAL) for comparison of fluorescence and target protein signals derived from different gels. Briefly, before gel electrophoresis, every protein sample was labeled with 2 µL of SMA basic L and 1 µL of Smart label working solution. Samples with of 10 µg of protein per sample, 12 µL of calibrator (CAL; 12.5, 25, and 80 kDa), and 5 µL of Biotinylated Protein Ladder (SeeBlue^TM^ Plus 2, Thermo Fisher Scientific) were applied to the gels. After conventional gel electrophoresis, the gel fluorescence of SMA basic L (signal: band at 80 kDa; channel: Alexa Fluor 488; gel loading control = GLO) and SMA label (signal: whole lane; channel: Cy5; gel total protein = GTO) were imaged. Following blotting, the blot fluorescence of the SMA label (signal: whole lane; channel: Cy5; blot total protein = BTO) and the blot chemiluminescent signal of the target protein (signal: band; chemiluminescence; band target protein = BTA) were detected. Finally, the normalization of gel load (based on SMA basic; GLO), normalization of Smart Label (based on SMA label), and normalization of target protein signal (based on total protein) were conducted using the SPL-normalization template following the manufacturer’s protocol. Results were expressed as SPL normalized target volume. Visualization and quantification were performed using the Chemidoc MP Imaging System and Image Lab^TM^ software version 6.0.1 (both from Bio-Rad).

### 4.5. Immunohistochemistry

Human liver specimens were harvested and fixed in 10% formalin overnight at 4 °C and embedded in paraffin. Hematoxylin and eosin (Thermo Fisher Scientific, Waltham, MA, USA) staining was conducted using a standard protocol on liver sections. Specifically, antigen retrieval was performed in 10 mM sodium citrate buffer (pH 6.0) by heating in a microwave on high for 10 min, followed by a 20-min cool down at room temperature. After blocking with 5% goat serum and the Avidin-Biotin blocking kit (Vector Laboratories, Burlingame, CA, USA), the slides were incubated with the anti-eIF4A1 primary antibody (1:100; ab31217, Abcam) overnight at 4 °C. Slides were then subjected to 3% hydrogen peroxide for 10 min to quench endogenous peroxidase activity and, subsequently, the biotin-conjugated secondary antibody was applied at a 1:500 dilution for 30 min at room temperature. Reaction detection was achieved using the Vectastain ABC-Elite Peroxidase Kit (Vector Laboratories, # PK-6100) with ImmPACT DAB (Vector Laboratories, SK-4105) as the chromogen. Slides were counterstained with Mayer’s hematoxylin.

### 4.6. Statistical Analysis

GraphPad Prism version 9.3.1 software (GraphPad Software Inc., La Jolla, CA, USA) and IBM SPSS version 26 software (IBM Corp., Armonk, NY, USA) were employed to evaluate statistical significance. The *p* values for TCGA data were obtained from the UALCAN analysis tool. The non-parametric Wilcoxon signed-rank test was used for paired sample comparison. Kaplan-Meier curves were evaluated using the log-rank test. For calculation of IC50s, data were transformed to log2, normalized, and non-linear regression (log)inhibitor vs. response–variable slope (four parameters) was performed. For the transformation of SEM to SD, SEM was multiplied by the square root of the sample size n. Tukey’s multiple comparison test was applied for multiple comparisons. Two-tailed values of * *p* < 0.5, ** *p* < 0.1, *** *p* < 0.01, and **** *p* < 0.001 were considered significant. Lowercase letters denoted statistical significance, as stated in the associated figure legends. All data are expressed as the mean ± SD or SEM.

## 5. Conclusions

Our results, obtained in silico and in vitro, constitute the initial step in an innovative direction to develop pharmacological strategies to treat liver cancer. This is particularly valuable owing to the lack of available tools for efficaciously treating advanced non-surgically resectable HCC. The fact that eIF4A1 was consistently upregulated in HCC in comparison with adjacent non-tumor liver tissue and the magnitude of this change was negatively correlated with the prognosis of HCC patients supports a role of eIF4A1 in HCC malignancy. This led us to evaluate the impact of eIF4A1 inhibitors on HCC cell viability. Interestingly, even at low nanomolar concentrations, these inhibitors triggered a remarkable reduction in proliferation and augmented apoptosis, paralleled by the inhibition of several oncogenic pathways. Moreover, the cytostatic effect of eIF4A1 inhibitors was further increased by the pan mTOR inhibitor, Rapalink-1.

During the last decade, considerable efforts have been made to characterize the molecular basis for the high refractoriness of HCC to currently available first- and second-line drugs [61]. This, together with the identification of new sensitizing targets, will permit us to overcome the current marked limitations in treating these patients, which in clinical practice means that they are subjected to treatments of low efficacy but with harmful side effects. In addition, unsatisfactory management delays the selection of alternative therapeutic options while allowing a Darwinian selection of the most resistant clones, which complicates the situation due to the development of cross-resistance.

In summary, the results of the present study highlight the pathogenic relevance of eIF4A1 in HCC and constitute an encouraging advance in developing new sensitizing strategies. Thus, the usefulness of combining eIF4A1 and mTOR inhibitors with currently used chemotherapeutic agents deserves further investigation.

## Figures and Tables

**Figure 1 ijms-24-02055-f001:**
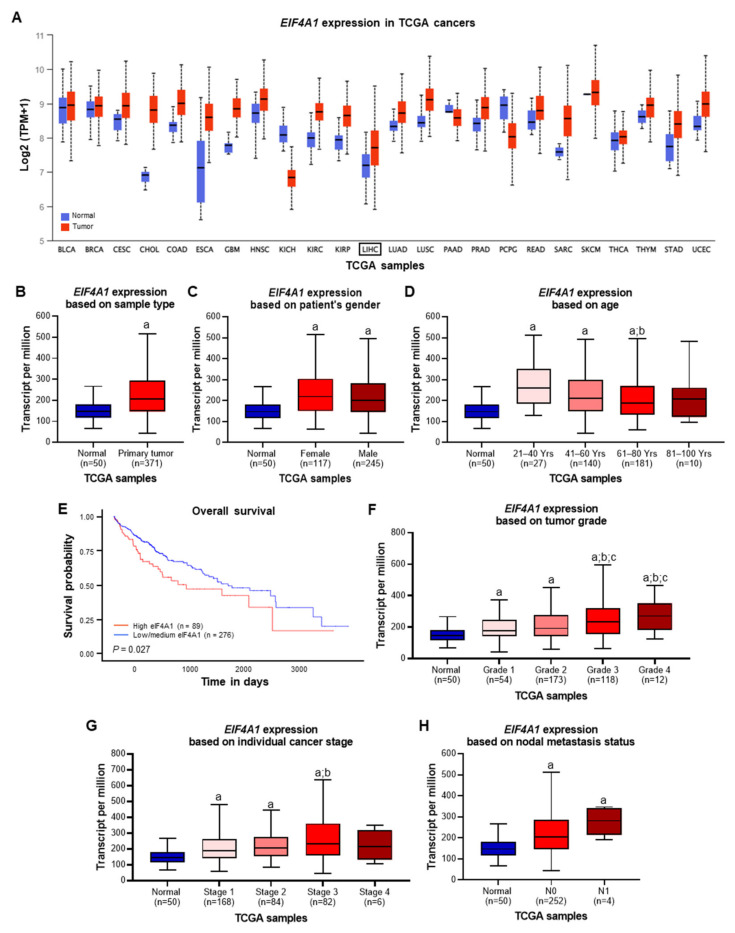
Biostatistical analysis from TGCA data of the *eIF4A1* gene in various tumor types and human hepatocellular carcinoma (HCC). Please refer to the main text for detailed explanations. (**A**) mRNA levels of *eIF4A1* in various tumor types, including liver hepatocellular carcinoma (LIHC). (**B**) mRNA levels of *eIF4A1* in non-tumorous (normal) and HCC (primary tumor) specimens. (**C**) mRNA levels of *eIF4A1* in HCC specimens based on gender. (**D**) mRNA levels of *eIF4A1* in HCC specimens based on age (Yrs: Years). (**E**) Kaplan-Meier curves in HCC patients based on *eIF4A1* mRNA levels. (**F**) mRNA levels of *eIF4A1* in HCC specimens based on tumor grade (grade 1: well-differentiated; grade 2: moderately differentiated; grade 3: poorly differentiated; grade 4: undifferentiated). (**G**) mRNA levels of *eIF4A1* in HCC specimens based on the individual cancer stage (stage 1: cancers localized to one part of the body; stage 2 and 3: locally advanced cancers; stage 4: spread to other organs and often metastasized cancers). (**H**) mRNA levels of *eIF4A1* in HCC specimens based on nodal metastasis status (N0: no evidence of cancer in regional lymph nodes; N1: cancer has spread to a single lymph node near the liver).

**Figure 2 ijms-24-02055-f002:**
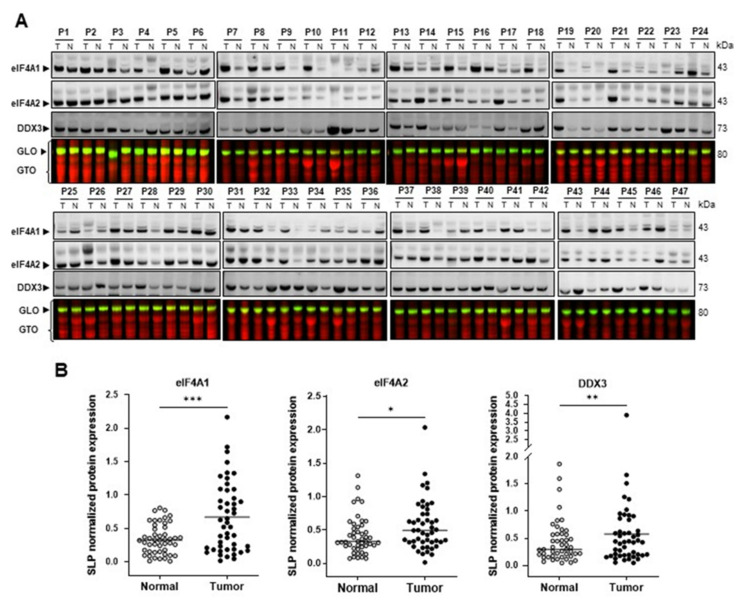
eIF4A1 is upregulated in human hepatocellular carcinoma (HCC; n = 47). (**A**) Levels of eIF4A1, eIF4A2, and DDX3 proteins in human HCC samples, as detected by western blotting using the SPL method. (**B**) Quantification of western blot band intensities. Dots representing the individual protein expression of one patient and their medians are shown (* *p* < 0.5; ** *p* < 0.1; *** *p* < 0.01; Wilcoxon matched-pairs signed rank test). Abbreviations: P, patient; T, tumor; N, non-tumor; GLO: Gel loading control (green); GTO: Total protein (red).

**Figure 3 ijms-24-02055-f003:**
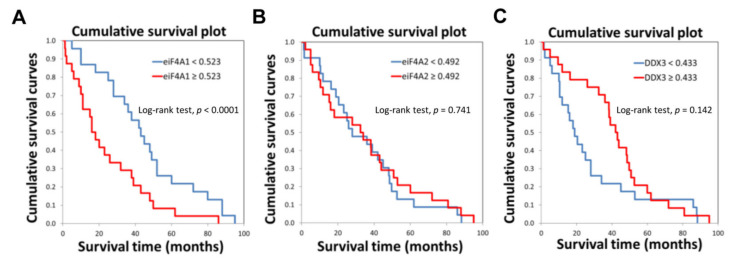
Levels of eIF4A1 protein negatively correlate with the survival of patients with hepatocellular carcinoma (HCC) (n = 47). Kaplan-Meyer curves in HCC patients for eIF4A1 (**A**), eIF4A2 (**B**), and DDX3 (**C**) proteins.

**Figure 4 ijms-24-02055-f004:**
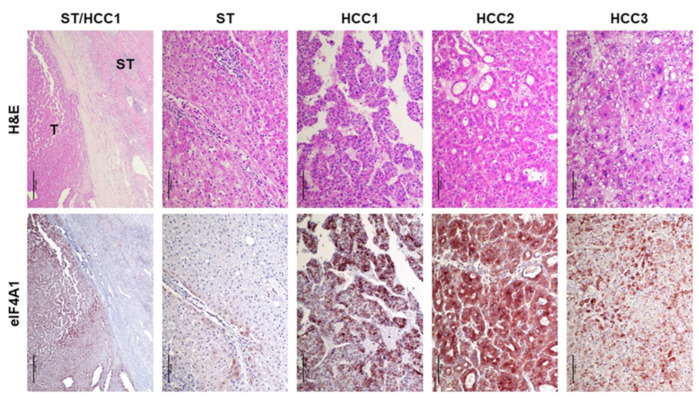
Representative immunohistochemistry patterns of eIF4A1 protein in human hepatocellular carcinoma (HCC; n = 356). The first left panels show an example of human HCC (denominated HCC1) at low magnification. Note the enhanced immunoreactivity for eIF4A1 in the tumorous part (T) compared with the neighboring non-tumorous surrounding tissue (ST), which exhibits faint/absent eIF4A1 staining. The second and third left panels depict higher magnifications for ST and HCC1. A second tumor (HCC2), characterized by intense, homogeneous eIF4A1 immunoreactivity, is shown in the fourth panel from the left. Finally, a third tumor (HCC3) in the right panels displays heterogeneous eIF4A1 immunolabeling. Abbreviation: H&E, hematoxylin and eosin staining. Original magnifications: 40× in the first left panels, 200× in all the other panels. Scale bar: 500 µm in the first left panels, 100 µm in all the other panels.

**Figure 5 ijms-24-02055-f005:**
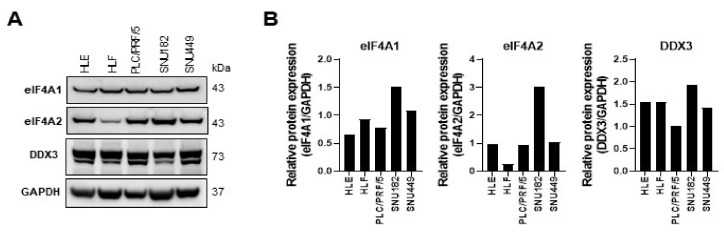
Human hepatocellular carcinoma cell lines express eIF4A1, eIF4A2, and DDX3 proteins. (**A**) The levels of the three proteins were assessed in HLE, HLF, PLF/PRF/5, SNU182, and SNU449 cell lines by western blot analysis. GAPDH was used as a loading control, and protein band intensities of eIF4A1, eIF4A2, and DDX3 were normalized to GAPDH levels (**B**).

**Figure 6 ijms-24-02055-f006:**
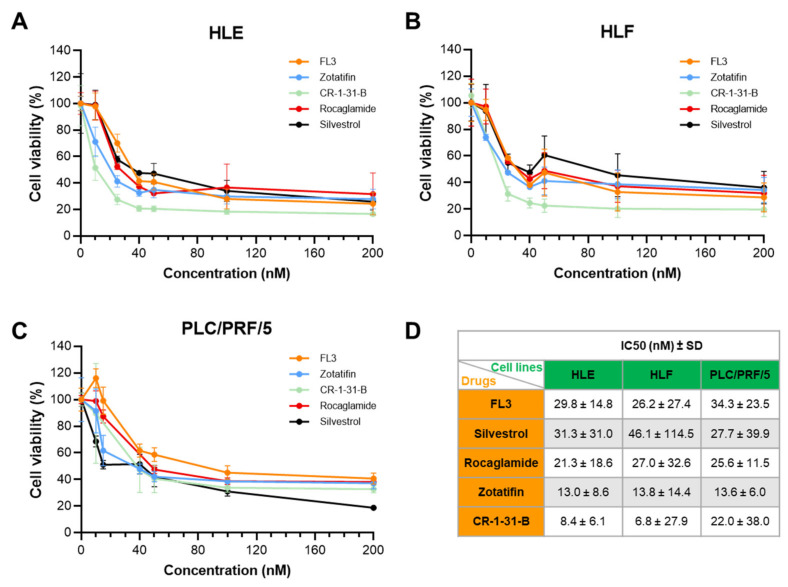
Cell viability of human hepatocellular carcinoma cell lines exposed to eIF4A inhibitors, as assessed by MTT assay. The eIF4A1 inhibitors, FL3, Zotatifin, CR-1-31-B, Rocaglamide, and Silvestrol, were administered to HLE (**A**), HLF (**B**), and PLC/PRF/5 (**C**) cell lines. Similar data were obtained in the SNU449 cell line (not shown). Data of two independent experiments with n = 4 technical replicates are represented as the percentage of DMSO-treated cells ± SD. A summary of the IC50 ± SD of the five drugs in the three cell lines is depicted in (**D**).

**Figure 7 ijms-24-02055-f007:**
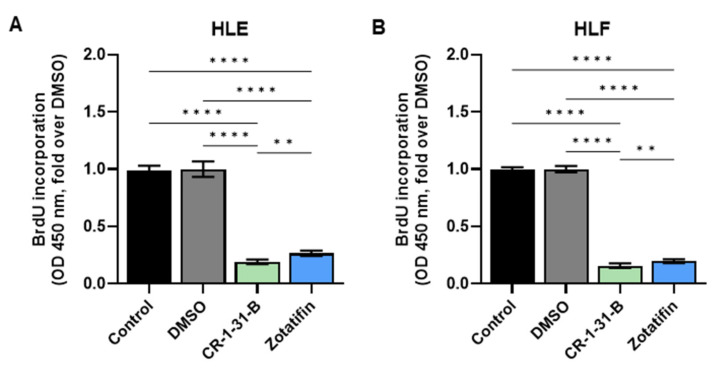
Effects of the eIF4A inhibitors, CR-1-31-B and Zotatifin, on the proliferation of hepatocellular carcinoma cell line monolayer cultures indicated by BrdU incorporation. BrdU incorporation assay was conducted on (**A**) HLE and (**B**) HLF cells treated for 48 h with 10 nM CR-1-31-B and 20 nM Zotatifin. Untreated cells and cells treated with DMSO served as controls. The fold-changes over DMSO of optical densities (OD) at 450 nm are presented. All results are expressed as mean ± SD of three independent experiments in triplicate. For statistical analysis, Tukey’s multiple comparisons test was performed (** *p* < 0.1; **** *p* < 0.001).

**Figure 8 ijms-24-02055-f008:**
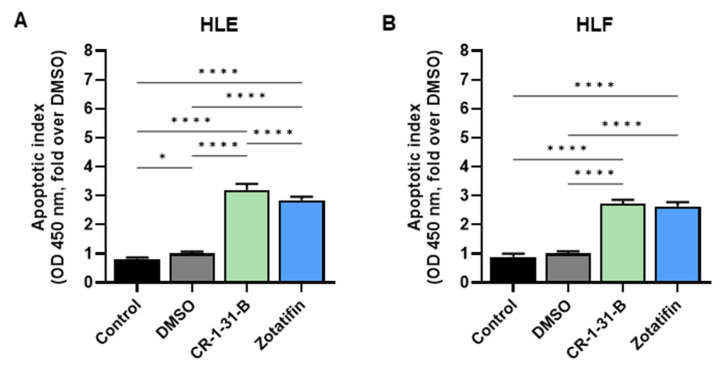
Effects of the eIF4A inhibitors, CR-1-31-B and Zotatifin, on the apoptosis of hepatocellular carcinoma cell line monolayer cultures. Cell death assay was conducted on (**A**) HLE and (**B**) HLF cells treated for 48 h with 10 nM CR-1-31-B and 20 nM Zotatifin. Untreated cells and cells treated with DMSO served as controls. The fold-changes over DMSO of optical densities (OD) at 405 nm are presented. All results are expressed as mean ± SD of three independent experiments in triplicate. For statistical analysis, Tukey’s multiple comparisons test was performed (* *p* < 0.05; **** *p* < 0.001).

**Figure 9 ijms-24-02055-f009:**
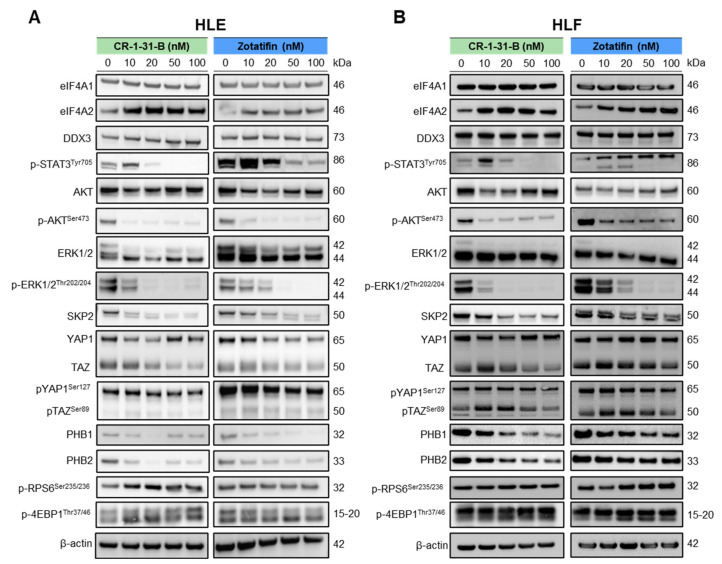
Effects of the eIF4A inhibitors, CR-1-31-B and Zotatifin, in HCC cell lines on the levels of elements of oncogenic pathways involved in hepatocarcinogenesis. Western blot analysis was used to assess the levels of several effectors of oncogenic cascades in HLE (**A**) and HLF (**B**) cell lines exposed to increasing concentrations of CR-1-31-B and Zotatifin inhibitors (0, 10, 20, 50, and 100 nM). Cells were treated for 48 h, and western blot analysis was conducted at this time point.

**Figure 10 ijms-24-02055-f010:**
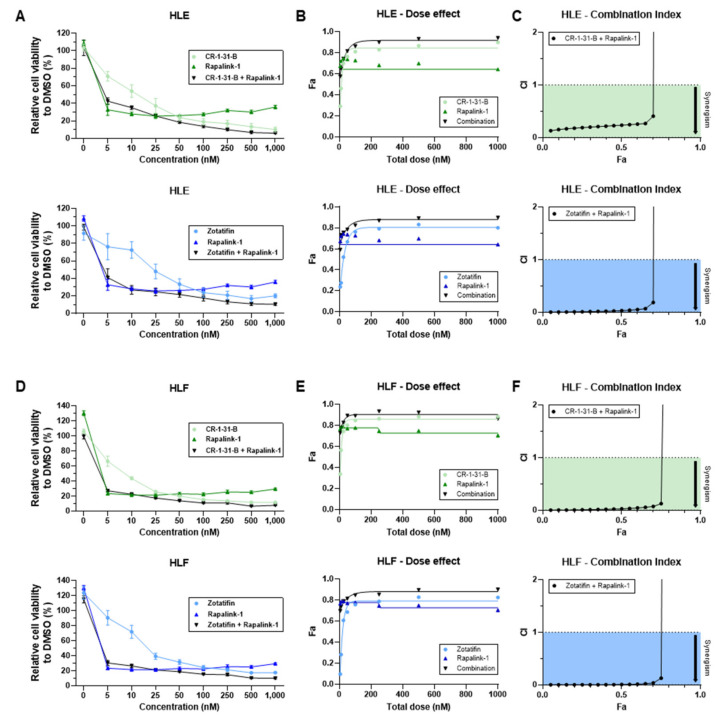
Highly synergistic cytotoxicity of combining the pan mTOR inhibitor, Rapalink-1, with CR-1-31-B and Zotatifin in hepatocarcinoma cell lines. (**A**) HLE and (**D**) HLF cells were exposed to various concentrations of compounds, alone or in combination, for 48 h. Cell proliferation based on the metabolic activity was assessed using the MTT assay and expressed as fold-change over DMSO control. The mean ± SEM of three independent experiments performed in technical triplicates are shown. (**B**,**E**) Dose effect curves of obtained fractions of dead cells (Fa) were used to determine the combination index (CI) (**C**,**F**) and synergism of drug combinations using CompuSyn software (https://www.combosyn.com, accessed on 1 November 2022) based on the Chou and Talalay method [50]. CI < 1 suggests synergism.

**Figure 11 ijms-24-02055-f011:**
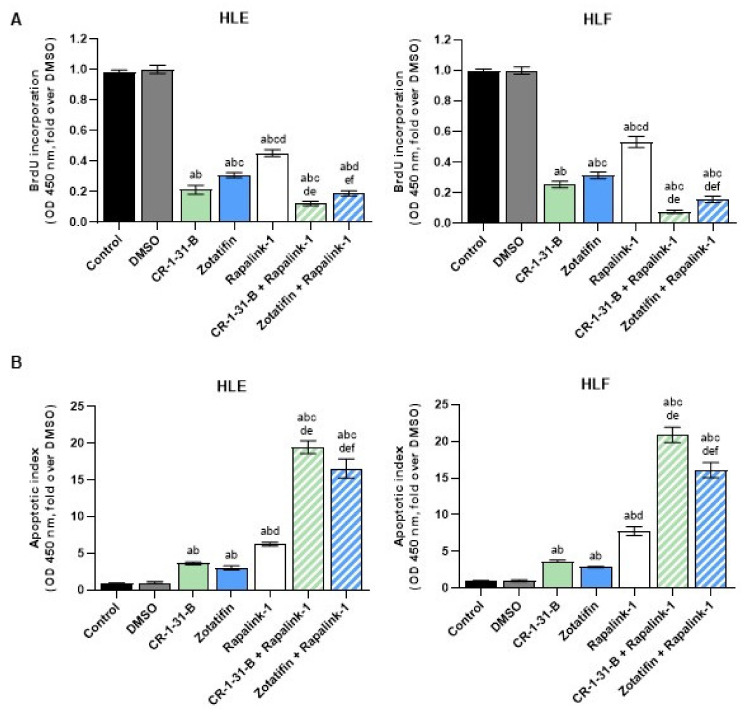
Combining the pan mTOR inhibitor, Rapalink-1, with CR-1-31-B and Zotatifin results in synergistic anti-growth activity in hepatocarcinoma cell lines. (**A**) HLE and HLF cells were exposed to the three drugs, alone or in combination, for 48 h and proliferation was assessed. (**B**) HLE and HLF cells were exposed to the three drugs, alone or in combination, for 48 h and apoptosis was determined. For the experiments in (**A**,**B**), the drugs were administered at their IC50 concentrations. All results are expressed as mean ± SD of three independent experiments in triplicate. For statistical analysis, Tukey’s multiple comparisons test was performed. Lowercase letters are used to denote statistical significance (a–f: *p* > 0.0001; a, vs. Control; b, vs. DMSO; c, vs. CR-1-31-B, d, vs. Zotatifin; e, vs. Rapalink-1; f, CR-1-31-B + Rapalink-1).

**Figure 12 ijms-24-02055-f012:**
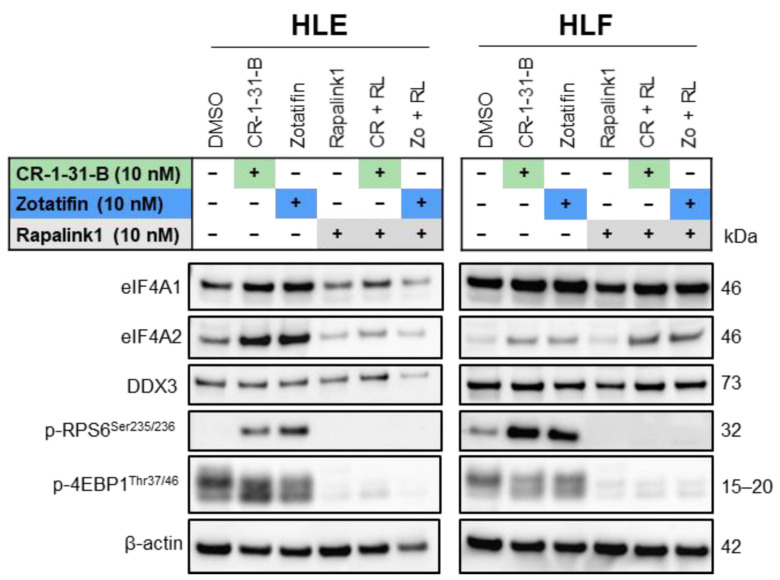
Complete inactivation of mTOR effectors after synergistic eIF4A and mTOR inhibition using CR-1-31-B and Zotatifin in combination with Rapalink-1 in HCC cell lines. Western blot analysis in HLE and HLF cell lines challenged with CR-1-31-B (10 nM) and Zotatifin (10 nM) and Rapalink-1 (10 nM) inhibitors, alone or in combination (1:1). Cells were treated for 48 h, and western blot analysis was conducted at this time point.

## Data Availability

Not applicable.

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
