# Peer review of "eIF4A1 Is a Prognostic Marker and Actionable Target in Human Hepatocellular Carcinoma"

_ijms, 2023, doi:10.3390/ijms24032055_

Round 1
Reviewer 1 Report
In the manuscript titled ‘eIF4A1 is a prognostic marker and actionable target in human hepatocellular carcinoma’, the authors described the potential effectiveness of targeting eIF4A1 as a therapeutic agent for Hepatocellular carcinoma.
In the introduction of the manuscript, the authors explained the rationale of targeting ribosome or translation machinery as an alternative way of targeting Liver cancer as these mechanisms are altered in cancer cells to enhance cancer-promoting phenotypes. Moreover, different inhibitors of eIF4A are also introduced as background to showcase the legitimacy of this target as described in other studies. The results and discussion of the findings are presented in ordered and eloquent manner; therefore the researchers are duly congratulated on their efforts.
The manuscript is sound and describes novel findings and can be recommended for publication in the International Journal of Molecular Sciences, provided a few minor comments are addressed.
Critique:
The description of samples should be given in the Figure legend as opposed to the results section (figure 1G, 1H) and only the findings should be discussed numerically or textually.
In line 220, eIF4A1 RNA is targeted, is this correct? Do the inhibitors used, target the RNA of eIF4A1? Please explain.
Why the cell lines used for treatments to measure viability were selected randomly? The cell lines exhibiting higher levels of eIF4A (figure 5) were not used ın Figure 6 data. Is there a reason pertaining to levels of eIF4A?
If eIF4A inhibitors don’t reduce the expression of eIF4A, is there any way to validate the action of these inhibitors on the activity of eIF4A other than the evidence given? such as ATPase or helicase activity?
In lines 310 and 313, instead of mentioning metabolic activity, the word ‘mitotic’ is written wrongly.
Figure 12 results should further be described and explained in light of relevant literature in the discussion of the manuscript.
Reviewer 2 Report
This manuscript by Steinmann et al represents an important validation of eIF4A1 as a prognostic marker for liver cancer. I recommend publishing after addressing the points below:
1. Error bars needs to be included in Fig. 6A0-C.
2. S.D. values should be added to IC50 values in Table D of Fig. 6.
3. Experimental details of using the Cell Proliferation Assay Kit should be clarified.
4. The information given regarding statistical analysis in the experimental section is not sufficient.
Reviewer 3 Report
Very interesting paper. I would suggest only to put the findings more in the context of the clinical applications. Which are the potential clinical applications of these findings? The authors should discuss this, also citing some references on the evolving clinical management of HCC (PMID: 33219585 and PMID: 31877664)
